# Deep Learning-Based Intelligent Forklift Cargo Accurate Transfer System

**DOI:** 10.3390/s22218437

**Published:** 2022-11-02

**Authors:** Jie Ren, Yusu Pan, Pantao Yao, Yicheng Hu, Wang Gao, Zhenfeng Xue

**Affiliations:** 1Intelligent Perception and Control Center, Huzhou Institute of Zhejiang University, Huzhou 313098, China; 2Institute of Cyber-Systems and Control, Zhejiang University, Hangzhou 310027, China; 3Science and Technology on Complex System Control and Intelligent Agent Cooperation Laboratory, Beijing 100191, China

**Keywords:** computer vision and its practical applications, robotics, deep learning, intelligent systems and control theory

## Abstract

In this research, we present an intelligent forklift cargo precision transfer system to address the issue of poor pallet docking accuracy and low recognition rate when using current techniques. The technology is primarily used to automatically check if there is any pallet that need to be transported. The intelligent forklift is then sent to the area of the target pallet after being recognized. Images of the pallets are then collected using the forklift’s camera, and a deep learning-based recognition algorithm is used to calculate the precise position of the pallets. Finally, the forklift is controlled by a high-precision control algorithm to insert the pallet in the exact location. This system creatively introduces the small target detection into the pallet target recognition system, which greatly improves the recognition rate of the system. The application of Yolov5 into the pallet positional calculation makes the coverage and recognition accuracy of the algorithm improved. In comparison with the prior approach, this system’s identification rate and accuracy are substantially higher, and it requires fewer sensors and indications to help with deployment. We have collected a significant amount of real data in order to confirm the system’s viability and stability. Among them, the accuracy of pallet docking is evaluated 1000 times, and the inaccuracy is kept to a maximum of 6 mm. The recognition rate of pallet recognition is above 99.5% in 7 days of continuous trials.

## 1. Introduction

Labor shortage and increasing labor cost are serious problems in today’s society. With the concept of Industry 5.0, it is imperative to promote industrial transformation and accelerate the automation and intelligent development of equipment in order to reduce the pressure brought by the rapid rise in labor costs, so more and more intelligent equipment is used in factories and storage environments [1,2,3]. Nowadays, the status of logistics equipment is increasing, and forklifts, as the main force of logistics handling equipment, have been widely used in many fields, such as factories, ports, and warehouses. However, as the requirements of the operating environment continue to increase, the handling equipment can no longer be operated by human hands, especially in special environments, such as high temperature, and hazardous and explosive environments. Along with the development of driverless technology, forklifts are also slowly approaching advanced technologies, such as intelligent identification, wireless transmission, and autonomous navigation and positioning. Intelligent forklifts can enhance the compound ability of forklifts, improve the overall operation level of forklifts, and gradually add more added value. Therefore, intelligent forklifts are the main development direction of forklifts in the future [2]. The operation of an intelligent forklift is quite straightforward; typically, it inserts and picks up pallets at one preset area before travelling to another to dump them off, accomplishing a full pallet transfer procedure. However, implementing such a straightforward approach presents numerous specific difficulties:Existing methods are more costly for determining whether a pallet is available at a certain location, while the recognition rate is low and also susceptible to interference by environmental factors [4].When the intelligent forklift inserts the pallet, there are problems of high implementation cost [5], low algorithm robustness and insufficient accuracy for the calculation of the relative position between the pallet and the forklift.When controlling the intelligent forklift to insert the pallet after the accurate position is calculated, a fixed control amount is usually used without considering the vehicle running state, which makes the control process deviate and eventually leads to errors in the inserting results [6].

We propose a deep learning-based intelligent forklift accurate cargo transfer system to address the aforementioned issues, as well as to increase the resilience and accuracy of the system. The system consists of various components with various sensors that cooperate to finish the pallet transfer operation. We specifically use RGB surveillance cameras to check whether there is any pallet that need to be transported at the pallet storage location. Once we determine that there are pallets, we send intelligent forklifts to the area. We then use the RGB-D (depth) camera that comes with the intelligent forklift to calculate the precise position of the pallets relative to the forklift. Finally, we use a high-precision control algorithm to control the forklift. The following three aspects make up the majority of the system features:To precisely determine whether there are pallets to be transported in the pallet storage area, we employ a Yolov5-based [7] pallet monitoring system and small target detection module, and its accuracy rate reaches more than 99.5%.To ensure that the final pallet insertion accuracy is within 6 mm, we calculate the real-time pallet position in relation to the intelligent forklift using the pallet position recognition system based on 3D Hough network [8].We present a high-precision tracing control approach for intelligent forklifts in order to increase the control accuracy, and the docking results obtained from 1000 experiments have an error of no more than 6 mm.

In our warehouse, we employ cameras to monitor pallets and intelligent forklifts to insert and remove pallets, as shown in Figure 1.

In the rest of the paper, we discuss related work in Section 2, describe our system in Section 3, then present experimental results in Section 4, and give conclusions in Section 5.

## 2. Related Works and Background

In this section, we discuss the work related to pallet monitoring, pallet position recognition, and intelligent forklift control.

### 2.1. Pallet Monitoring

When intelligent forklifts are first put into use, it is usually the human who determines whether there are pallets to be inserted and picked up at a specific point. Workers usually hold devices, such as tablet computers or pagers, and send commands to the intelligent forklifts to insert and pick up pallets. However, this approach necessitates human involvement, which wastes labor and is ineffective.

In some projects, the use of sensor-assisted automatic identification techniques has begun [9]. In such projects, sensors are typically installed at pallet storage locations to detect the presence or absence of pallets [10], and the results are then transmitted to the dispatching system via a network cable so that the intelligent forklifts can determine whether a pallet needs to be moved at a specific location. However, using this approach in a large-scale storage environment requires the deployment of sensors at each pallet storage location, which greatly increases the difficulty and cost of implementation.

Recent studies have looked into using RGB surveillance cameras to detect the presence or absence of pallets [11]. This recognition is typically based on the conventional image recognition scheme, which assumes that the appearance of the pallet is mostly visible and clearly distinguished from its surroundings, and then extracts the pallet from the image using techniques such as image segmentation, template matching, etc. This method does not account for the fact that a pallet may have goods covering all or nearly all of its surface, which makes it easy to mistake a pallet for nothing because the camera cannot gather enough data on the pallet’s color and contours.

In 2021, Joo et al. proposed a Yolov3-based pallet recognition method [12] which is designed for the industry and can recognize pallets more steadily than conventional image recognition techniques. However, the technique necessitates that the camera be placed in close proximity to the pallet in order to collect data. One camera cannot effectively monitor a large area of pallets, and the recognition rate is low for pallets with a small pixel share.

### 2.2. Pallet Position Recognition

Pallet position recognition relative to forklift has always been an industry challenge. The pallet must frequently be placed manually or mechanically at the exact location (error less than 1 cm) on the shelf during the initial stages of unmanned forklift use. The intelligent forklift only needs to get to the fixed position each time to finish inserting and retrieving the pallet in this situation because the position of the pallet and the shelf is essentially fixed. This method is not suitable for the automated operation of the plant because it requires too much accuracy in pallet placement, and if there is a mistake, it is easy to happen that the forklift cannot insert the pallet and needs manual assistance.

A technique for using auxiliary markers, such as QR codes, for position recognition has surfaced in the industry as a solution to such issues [13]. The pallets are marked with additional markers, and an on-board scanning gun is used to find the markers. Because it can calculate the position of the pallet in relation to the forklift based on the location of the QR code while entering the pallet information, it has been widely used in the industry. However, we prefer a method that identifies pallets based on their own shape, texture, and other information rather than methods that require markings to be posted on each pallet, which requires a lot of work in the pre-deployment stage.

Garibotto et al. [14,15] proposed a vision-based algorithm to detect the central hole of a pallet, where the hole features of the pallet are extracted after a pre-segmentation of the image, and then the geometric model of the pallet is projected onto the image plane for position estimation. However, the traditional image approach used by this method to identify the position of the pallet hole makes it more susceptible to interference. This is especially true if the shape of the portion of the goods above the pallet is similar to the shape of the hole position, which is typically a simple rectangle.

### 2.3. Forklift Precision Control Algorithm

The more common forklift control algorithm is based on the improvement of PID control [16], and the desired control effect is achieved by adjusting different control parameters according to the actual usage. However, due to its ease of use and constrained range of adjustment, this method is difficult to adapt to complex factory or warehouse environments and frequently exhibits the trait of low control accuracy in use.

Jiang, Zhizheng et al. proposed a Robust H-based forklift control algorithm [17] to model the dynamics of an electric power steering (EPS) system of an electric forklift. The standard H control model of the EPS system is transformed to derive the generalized equation of state for the EPS of an electric forklift. The principle of genetic optimized robust control is described, and the constraint function of genetic algorithm (GA) is constructed for the parameter optimization of the weighting function of the H control model, and the genetic optimized robust controller is derived. The accuracy and robustness of the forklift control process are effectively enhanced. This method, however, also only focuses on the function of EPS in forklift control and neglects to design for the actual operating circumstances during forklift operation or take into account the dynamic adjustment of the control volume to increase control accuracy.

## 3. Intelligent Forklift Cargo Precision Transfer System

In a factory or warehouse, intelligent forklifts are needed to accurately insert and transport pallets, as well as autonomously determine whether there is any pallet in the pallet storage area that need to be transferred. We introduce an intelligent forklift cargo precision transfer system to carry out this function, as depicted in Figure 2. In order to determine whether there is any pallet that need to be transported at the pallet storage location, we use a standard RGB surveillance camera. The forklift is then dispatched to the area of the pallet, and the exact position of the pallet is recognized by the RGB-D camera that comes with the forklift. Finally, according to the recognized exact position, a high-precision control algorithm is used to control the forklift to insert and pick up the pallets.

### 3.1. Pallet Monitoring Module

Pallet monitoring is the cornerstone of pallet transfer by forklift, and we carry out this monitoring task using standard RGB monitoring cameras. One advantage of these RGB cameras is that they are less expensive than photoelectric sensors or RGB-D cameras, and they are also much easier to deploy due to the fact that one RGB monitoring camera can cover at least 50 pallet positions. To prevent the influence of items placed on the pallets on the recognition outcomes, we used the side aperture feature of the pallets, which is not easily covered up, as the identification mark. We used a Yolov5-based network for pallet monitoring. However, the size of the pixels occupied by the same pallet in the same image is different because of the viewing angle when the surveillance camera is placed. The traditional Yolov5 scheme will easily fail when the pallet is far from the camera because the pallet’s pixels in the field of view will be small. In order to handle these situations, we added a small target detection module and a convolutional block attention module (CBAM).

#### 3.1.1. BackBone Modules

CspDarknet53 [18] is a Backbone structure based on Darknet53 [19], which contains 5 CSP modules. Each CSP module’s convolution kernel is 3 × 3 in size with stride = 2, so it can function as a downsampling. The input image is 608 × 608, and since Backbone has 5 CSP modules, the pattern of feature map change is 608 → 304 → 152 → 76 → 38 → 19. After 5 CSP modules, the 19 × 19-inch feature map is obtained.

#### 3.1.2. Neck Module

The feature extractor of this network uses a new enhanced bottom-up pathway FPN [20] structure that improves the propagation of low-level features. Each stage of the third pathway takes the feature map of the previous stage as input and processes them with a 3 × 3 convolutional layer. The output is added to the feature maps of the same stage of the top–down pathway via lateral connections, and these feature maps inform the next stage. Adaptive feature pooling is also used to recover the corrupted information path between each candidate region and all feature levels, aggregating each candidate region at each feature level to avoid arbitrary assignment.

#### 3.1.3. Small Target Detection

We incorporate a transformer prediction head (TPH) into Yolov5 in order to detect small targets. A low-level, high-resolution feature map, which is more sensitive to small objects, is used to generate the added TPH. We also swap out some convolution and CSP blocks with the transformer encoder block. The transformer encoder block in CSPDarknet53 has more information acquisition advantages than the original bottleneck block. The first sub-layer in each transformer encoder block is a multi-headed attention layer, and the second sub-layer (MLP) is a fully connected layer. Each sub-layer is connected by residuals. The transformer encoder block improves the ability to record various local details and can also make use of the self-attentive mechanism to unlock the potential of feature representation.

#### 3.1.4. Convolutional Block Attention Module (CBAM)

CBAM [21] is a simple but effective attention module. It is a lightweight module that can be integrated into a CNN and can be trained in an end-to-end manner. According to the experiments in the paper [22], the performance of the model is greatly improved after integrating CBAM into different models for different classification and detection datasets, which proves the effectiveness of the module. In images of pallet surveillance, large coverage areas can contain a high number of interference terms. Using CBAM can extract attention regions and help the network resist confusing information and focus attention on useful target objects.

### 3.2. Pallet Positioning Module

The relative poses between the target pallet and the intelligent forklift must be identified after the forklift approaches the target pallet in order to provide precise real-time pose for the forklift’s control algorithm. In order to accomplish this, we suggest a novel approach based on a deep 3D Hough voting network. To be more specific, we employ a RGB-D camera to record the color and depth data of the pallet. We then input the color and depth data into a feature extraction module to extract the surface features and geometric information of the pallet. This extracted information is then fed to a key point detection module $M_{k}$ to predict the offset of each point relative to our specified key points, which are typically defined as the 8 corner points of the two apertures of the pallet. Additionally, we employ a center voting module $M_{c}$ to predict each point’s offset from the target center and an instance segmentation module, $M_{s}$, to predict the label of each point. Finally, the obtained 8 key points are used to estimate the pallets’ poses relative to the forklift by the least squares method. The whole algorithm flow is shown in Figure 3.

#### 3.2.1. Key-Point Detection Module

The pallet image is fed to the key-point detection module $M_{k}$ after feature extraction to detect key-points on the pallet. The main function of $M_{k}$ is to predict the Euclidean translation offset from the visible points to the key-points, and these visible points and the predicted translation offset work together to finally be able to vote out the key-points. These voted points are pooled together by a clustering algorithm [23], and the center of the pool is taken as the final key point.

The loss function of $M_{k}$ is calculated as follows. Given a set of extracted feature points ${\left\{p_{i}\right\}}_{i=1}^{N}$, where $p_{i}=\left[x_{i},f_{i}\right]$, $x_{i}$ denotes the 3D coordinates of the points and $f_{i}$ denotes the point features. Similarly ${\left\{kp_{j}\right\}}_{j=1}^{M}$ is used to denote the selected key points. We use ${\left\{of_{i}^{j}\right\}}_{j=1}^{M}$ to denote the translation offset of the *i*-th point with respect to the *j*-th key point. Thus, the key point can be represented as $kp_{i}^{j}=x_{i}+of_{i}^{j}$. To supervise the learning of $of_{i}^{j}$, we use the loss function:(1)$$L_{\mathrm{key}-\mathrm{points}\;}=\frac{1}{N}\sum_{i=1}^{N}\sum_{j=1}^{M}∥of_{i}^{j}-of_{i}^{j*}∥$$

$of_{i}^{j*}$ is the true value of the translation offset. *M* is the number of selected key-points, which is usually selected as 8 in the system. *N* is the number of feature points.

#### 3.2.2. Instance Segmentation Module

In order to handle the case where there are multiple objects in the image, i.e., multiple goods or other objects in addition to the pallet, it is necessary to segment the other objects from the pallet in order to extract the key points more accurately. As a result, we present the $M_{s}$ instance segmentation module. The instance segmentation module $M_{s}$ predicts the semantic label of each point using the extracted point-by-point data. To supervise the learning of this subject, we employ Focal Loss [24].
(2)$$L_{s}=-\alpha {\left(1-q_{i}\right)}^{\gamma }\log\left(q_{i}\right),\;\mathrm{where}\;q_{i}=c_{i}\cdot l_{i}$$
with α the α-balance parameter. γ the focusing parameter. $c_{i}$ the predicted confidence for the *i*-th point belongs to each class and $l_{i}$ the one-hot representation of ground true class label.

We also employ a center voting module, $M_{c}$, to help distinguish between the various instances. We use a similar module to CenterNet [25], but expand the center points from 2D points to 3D points. This module is able to predict the Euclidean translation offset of each point to its object center in order to achieve a better instance segmentation. Similar to the key-point detection module mentioned above, it can be used to regard the object’s center as a certain kind of key-point. We denote by $\Delta x_{i}$ the translational offset of each feature point with respect to its object center, then the learning of $\Delta x_{i}$ can be supervised by the following loss function.
(3)$$L_{c}=\frac{1}{N}\sum_{i=1}^{N}∥\Delta x_{i}-\Delta x_{i}^{*}∥$$
where *N* denotes the total number of seed points on the object surface and $\Delta x^{*}$
*i* is the ground truth translation offset from seed $p_{i}$ to the instance center. It is an indication function indicating whether point $p_{i}$ belongs to that instance.

#### 3.2.3. Network Architecture

As shown in Figure 3, the first part of the network is a feature extraction module. In this module, a PSPNet [26] is used to extract the appearance information in RGB images. PointNet++ [27] extracts the geometric information in the point cloud and its normal mapping. The two are fused by the DensionFusion block [28] to obtain the combined features of each point. The next $M_{k}$, $M_{s}$ and $M_{c}$ consist of shared multilayer perceptrons (MLPs). We supervise the learning of module $M_{k}$, $M_{s}$ and $M_{c}$ jointly with a multi-tasks loss:(4)$$L_{multi-task}=\lambda_{1}L_{k}+\lambda_{2}L_{s}+\lambda_{3}L_{c}$$

We sample *n* = 12,288 points for each RGB-D image frame and set $\lambda_{1}=\lambda_{2}=\lambda_{3}=1.0$.

#### 3.2.4. Least Squares Fitting

In the least-square fit [29], we denote the final set of 8 key points inferred by the network as ${\left\{kp_{i}\right\}}_{j=1}^{M}$. The coordinates of this point set are in the camera coordinate system, and the corresponding set of 8 points in the pallet coordinate system is denoted as ${\left\{kp_{i}^{\prime }\right\}}_{j=1}^{M}$. To obtain the bit-pose relationship (R,t) between the pallet and the camera, we will minimize the set of these two points between the following loss functions.
(5)$$L_{least-squares}=\sum_{j=1}^{M}{∥kp_{j}-\left(R\cdot kp_{j}^{\prime }+t\right)∥}^{2}$$

The camera and the intelligent forklift are often fixedly coupled when in use. After obtaining the relative position of the pallet and the camera, only one step of external reference conversion needs to be added to obtain the relative position between the forklift and the pallet.

### 3.3. High Precision Control Module

Once the pallet position has been determined, we must accurately control the forklift to insert the pallet in that position. Traditional forklift control models [30] and algorithms are overly simplistic, especially the actual running condition of the vehicle and the amount of dynamic adjustment control are not considered enough, which frequently results in significant inaccuracies when used in practice. We propose a high-precision trajectory control method for intelligent forklifts that incorporates forklift motion cycle prediction into the control process, continuously updates the intelligent forklift prediction model, and determines the optimal control amount in order to achieve the goal of increasing control accuracy.

In order to describe the motion of the intelligent forklift in the prediction cycle, a discrete prediction model based on the forklift model is required. Using the forklift motion model as the base model, a prediction model for the non-linear optimization problem is established and a discrete vehicle model in incremental form is performed, as shown:(6)u(k)=u(k−1)+Δu(k)

As shown in Figure 4, according to the recurrence relationship of the incremental model, the *i*-th future prediction state can be represented by the initial state and the sequence of *i* control increments to complete the construction of the non-linear prediction model.

Combining the path tracking objective function based on the approximate tracking error [22], the non-linear prediction model and constraints [31], the path tracking optimization problem based on the approximate tracking error is constructed as shown:(7)$$\min J=\sum \left({∥e∥}^{2}+{∥\Delta u∥}^{2}+{∥u∥}^{2}\right)$$

Additionally, the constraints are added to the optimization problem to obtain the prediction state and control increment by minimizing the sum of squares of approximate tracking error, control increment, and control quantity in the prediction cycle to complete the update of the model and control quantity.

In general, we update the prediction model based on the observed values and the discrete prediction model, establish the path tracking objective function based on the approximation error, obtain the optimal control volume, and combine the previous moment control volume output to the controlled vehicle to complete the high-precision tracking control.

## 4. Experiment Results

To guarantee that the findings obtained in this article are consistent with the results in actual use, we run the system through real-world scenarios in order to obtain more realistic and accurate results.

### 4.1. Experiments Environment Construction

The whole system consists of three main parts: pallet monitoring module, pallet positioning module, and high-precision control module. The supporting scheduling system is not discussed in any detail in this work. The forklift body control algorithm, or the high-precision control module, does not need to build an experiment environment and can be tested directly with the forklift.

#### 4.1.1. Pallet Monitoring Module

The monitoring system consists of 8 network RGB cameras and a dispatch server, and 8 network cameras are connected to the server through a switch. We placed 53 pallets in the field of view of camera 8-th and multiple pallets in different locations and numbers in the field of view of other cameras, while placing goods on the pallets to simulate real scenarios. We expect the algorithm to have a good recognition rate for this complex and variable situation. Figure 5 shows the working situation of the monitoring system.

#### 4.1.2. Pallet Positioning Module

A RGB-D camera was used for pallet position recognition, which was installed in a fixed position on the forklift and the data were uploaded to the forklift mounted IPC for position calculation via a network cable. In order to avoid the fork tines of the forklift from appearing in the image, we adjusted the camera to a suitable position so that it could see the complete pallet hole position without seeing the fork tines of the forklift. Figure 6 shows the RGB-D camera and its recognition results.

### 4.2. Experiment Results

#### 4.2.1. Pallet Monitoring Module

In order to test the monitoring effect of pallets from multiple cameras, we placed pallets at different locations in the 8-way camera field of view for actual testing. Considering that pallets are likely to be stacked and loaded with goods at customer sites, we placed 2 and 3 layers of pallets in a large number of pallets and placed goods at the top edge of the pallets. We used the side aperture feature of the pallets, which is not easily concealed, as the identification mark in order to prevent the influence of the goods on the pallets on the recognition results. The placement of the pallets under each camera is shown in Figure 7.

The figure shows how the pallet monitoring system can accurately present the results of pallet recognition for multiple scenarios. Our algorithm places a red box in the side hole position of the recognized pallet to show that a pallet was identified. To see the recognition results more clearly, we use the camera 8-th with the highest number of pallets for analysis.

Before adding the small target detection and CBAM described in 3-A, the recognition effect of camera 8-th is shown in Figure 8. It is obvious that some pallets located further away from the camera cannot be detected, which is typically caused by unreliable recognition brought on by low pixels.

We have used small target detection and CBAM to solve such problems, and the recognition effect after addition is shown in Figure 9. It can be clearly seen that the unrecognized pallets in Figure 8 have been recognized in Figure 9, and all unobstructed pallets in the field of view are recognized.

To better reflect the advantages of our algorithm, we list the comparison of existing algorithms with our algorithm in Table 1.

As can be seen from the table, the implementation cost of the photoelectric sensor method is high; the traditional image method is susceptible to interference; and the deep learning method based on Yolov3 does not have a high recognition rate for small targets. Although our method, with low cost, still has a high recognition rate for small targets.

In order to test the reliability of the system operation, we designed the experiment for continuous 7 × 24 h non-stop testing. We adjust the position of multi-layer pallets and goods within the scene before the start of each day’s experiment to ensure that the conditions are different each time. Following the change, we will count the unobstructed pallets and enter that number as a parameter in the system. The system will calculate the recognition rate by comparing the pallets it has identified with the input parameter. Table 2 shows the number of pallets and the recognition rate during the 7-day experiments.

As can be seen in Table 2, the recognition rate is less than 100%, indicating that there are still cases where individual pallets are not recognized. The recognition rate exceeds 99.5% for about 70 pallets tested continuously for 7 days, including single-layer, multi-layer, with or without goods. For the situation where a large number of pallets appear under one camera, recognition can still be performed stably as long as the pallets are not blocked from each other before (i.e., a reasonable distance between pallets is set). The eight cameras are able to output data streams together for simultaneous recognition of pallets in the area.

#### 4.2.2. Pallet Positioning Module and High Precision Control Module

In the pallet position recognition module, we use RGB-D camera to collect data from several different positions and angles of the pallet for calculation, and provide the results to the intelligent forklift to control the forklift for precise pallet insertion. Since we expect to obtain the overall accuracy of pallet insertion by the forklift, we do not distinguish the positional accuracy from the control accuracy; instead, we judge the ultimate total insertion accuracy.

We collect a lot of data in order to ensure that the findings of our trials covered the majority of usage cases. Given that the relationship between forklifts and pallets is typically not fixed before pallets are inserted, we set up a variety of starting positions: the distances between the pallet hole plane and camera plane are 1 m, 2 m, and 3 m; the horizontal distances are 0 m, −0.2 m, and +0.2 m; and the relative angles between the camera and pallet are 0 deg, +15 deg, and −15 deg. Additionally, we established two separate heights, with the pallet placed on the ground being recorded as 0, and the pallet placed on a shelf that is 40 cm high being recorded as 0.4 m. This is because a single pallet may be placed on either the ground or a shelf. We carried out pallet docking approximately 10 times for each scenario, comparing the inaccuracy of each result with the initial result. In other words, we completed a total of 1000 dockings, of which 550 were for the 1-layer pallets, 180 were for the 2-layer pallets, and 260 were for the 3-layer pallets.

We have placed the pallets in different positions and layers and recorded the final error. We recorded the error value of each result compared with the first result and drew it in Figure 10, where the x-axis represents the number of docking times and the y-axis represents the result error value.

Figure 10 shows that the accuracy in both the x-direction and the y-direction can be controlled to about 5 mm. However, in some circumstances, particularly when dealing with three-layer pallets, the error will increase to 6 mm because multi-layer pallets produce more interference than single-layer pallets.

Overall, our pallet docking accuracy can be controlled within ±6 mm and the coverage range is from 1–3 m and from −15° to +15°. When compared to the measures in the literature [32], the coverage range is roughly the same, but it does not test for severe pallet angle deflection. More importantly, the pallet accuracy in the literature [32] is only ±3 cm, which is much lower than our ±6 mm accuracy. Similar coverage and coverage angle experiments to ours were carried out in the literature [33], but their maximum error of recognition was 10.5 mm after only 135 experiments, which was 75% higher than our maximum error of 6 mm. Its mean error and standard deviation are also significantly higher than ours. Meanwhile, our number of experiments was 7.4 times higher than his, and we measured the final error after pallet docking, which is the accumulation of recognition error and control error values, indicating that our identification error values are lower. On the other hand, the literature [33] used two sensors to achieve the results expressed in the article, while we obtained better results by using only one RGB-D camera. The experiment results are shown in Table 3.

The data in the table shows that the maximum error of the existing algorithm for pallet position recognition can be 1–3 cm, which is a risk that the fork tines cannot enter the pallet hole for the high-precision pallet insertion action of the forklift. Our module is able to reduce the maximum error to 6 mm, which fully meets the needs of practical use. At the same time, our system requires only one RGB-D camera to collect data without the assistance of other sensors, which is much less costly.This demonstrates that our approach outperforms the majority of systems now used in industrial settings in terms of accuracy, resilience, and cost.

According to the above experimental results, the recognition rate of our pallet monitoring system has been significantly improved after the addition of small target detection, and the recognition rate reached 99.5% under 7 days of continuous experiments; our position recognition and control algorithms work together to control the forklift interpolation accuracy within 6 mm in 1000 experiments, which fully meets the requirements of accuracy in practical use and outperforms existing algorithms in both coverage and accuracy.

## 5. Conclusions

We propose an intelligent forklift cargo accurate transfer system, which consists of three main parts: pallet monitoring module, pallet positioning module, and high-precision control module. The system creatively introduces small target detection into pallet recognition to improve the recognition rate; using Yolov5-based pallet pose detection algorithm to improve the coverage and recognition accuracy of the algorithm. For the whole system, we proved its effectiveness and reliability by actually collecting a large amount of data. Among them, the recognition correct rate of pallet monitoring module reaches more than 99.5% in 7 days continuous experiments, and the insertion error of the intelligent forklift is below ±6 mm after 1000 experiments through pallet positioning and control algorithm. The performance of the whole system is greatly higher than the existing common systems, and has been used in factories and warehousing environments on the ground. We will further investigate the use of lightweight networks to refine our system in order to reduce computational resource consumption and computation time. In this way, the cost of commercializing the system can be further reduced.

## Figures and Tables

**Figure 1 sensors-22-08437-f001:**
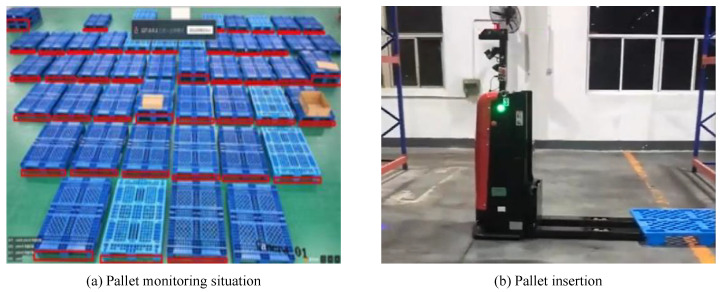
System operation diagram. We have constructed the entire system that is detailed in this paper in the warehouse. One of the eight cameras in the pallet monitoring system, which can monitor the presence of pallets in the storage area and mark them with red boxes when they are found, is illustrated in (**a**). The intelligent forklift arrives at point (**b**), determines the location of the pallets, and then executes the insertion and extraction procedure depicted in the figure after realizing that the pallets need to be moved.

**Figure 2 sensors-22-08437-f002:**
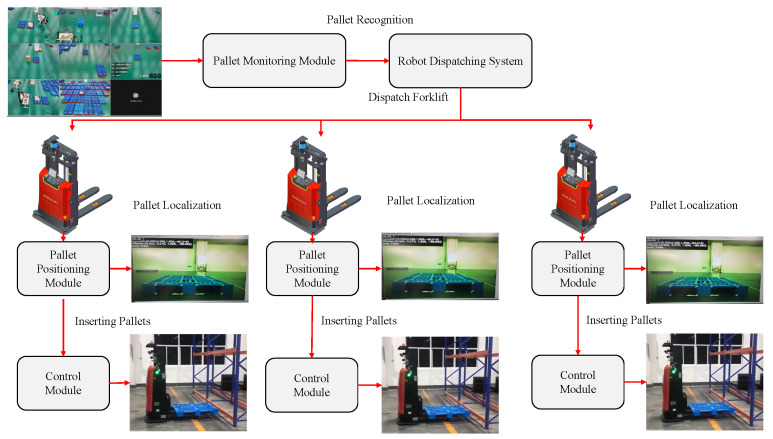
The overall flow of the intelligent forklift precision cargo transfer system. RGB camera captures images of the pallet storage area and transmits them to the pallet monitoring module, which identifies the pallets and informs the dispatching system. The dispatching system dispatches forklifts to insert and pick up the identified pallets. The forklift reaches the pallet and uses the pallet positioning module to identify the pallet position, and after the position is identified, it is handed over to the high precision control module to control the vehicle to insert and pick up the pallet.

**Figure 3 sensors-22-08437-f003:**
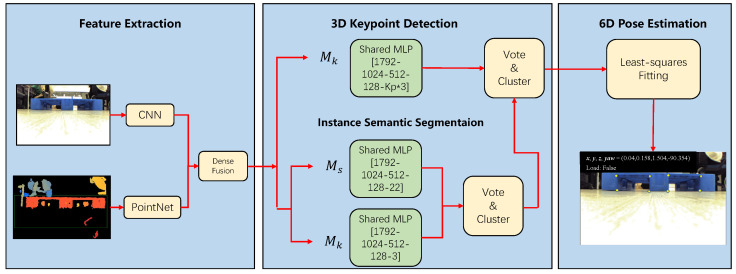
Flow of pallet position recognition algorithm. To predict the translational offsets to the key-points, center points, and labels of each point, the feature extraction module extracts features from the RGB-D images. These features are then fed into the $M_{k}$, $M_{s}$, and $M_{c}$ modules. The key-points in the same instances are then voted to the key-points of the instance they belong to after a clustering algorithm is used to distinguish between the various instances. Finally, a least squares method is used to fit the bit pose based on the eight extracted key points. $K_{p}$ is the number of key points.

**Figure 4 sensors-22-08437-f004:**
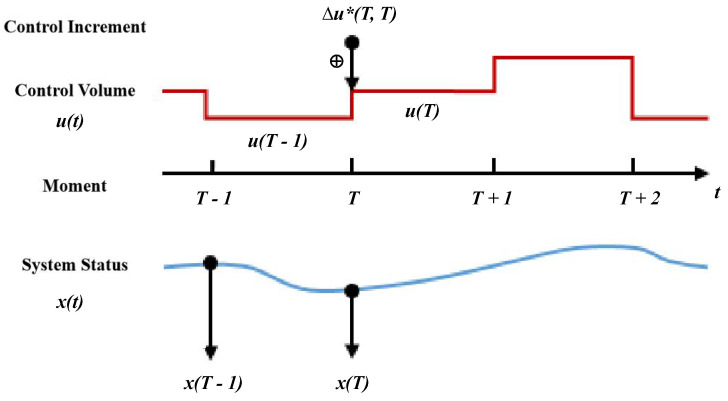
Incremental model recurrence relationship. Get the value at different times.

**Figure 5 sensors-22-08437-f005:**
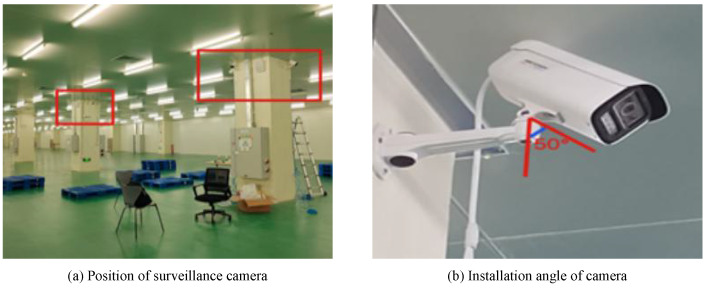
Pallet monitoring deployment. The surveillance cameras were mounted as high as possible and angled so that more pallets could be monitored. The network camera was installed at a height of approximately 3.27 m and the network camera was overhead (with vertical lines) at an angle of approximately 50 deg.

**Figure 6 sensors-22-08437-f006:**
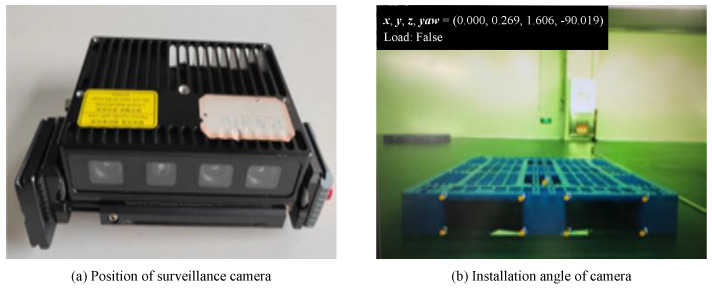
Pallet positioning module deployment. An on-board RGB-D camera was used and the camera was adjusted to a suitable position. The 8 corner points of the pallet aperture are recognized and the result of the pose calculation is shown in the upper left corner of the image.

**Figure 7 sensors-22-08437-f007:**
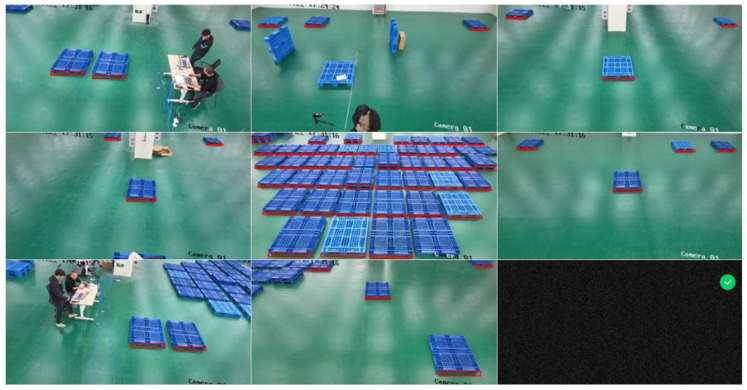
Pallet placement under each camera. Different number of pallets with different number of layers were placed under different cameras and tested repeatedly.

**Figure 8 sensors-22-08437-f008:**
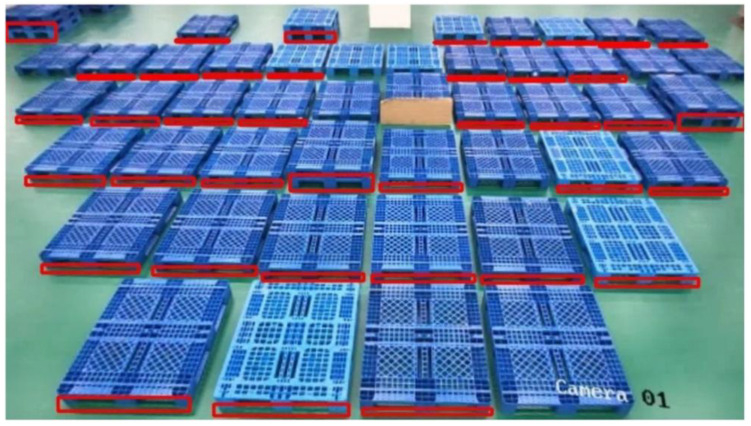
The recognition effect of camera 8-th. The majority of the pallets in the field of view are identified, however certain specific pallets are still not.

**Figure 9 sensors-22-08437-f009:**
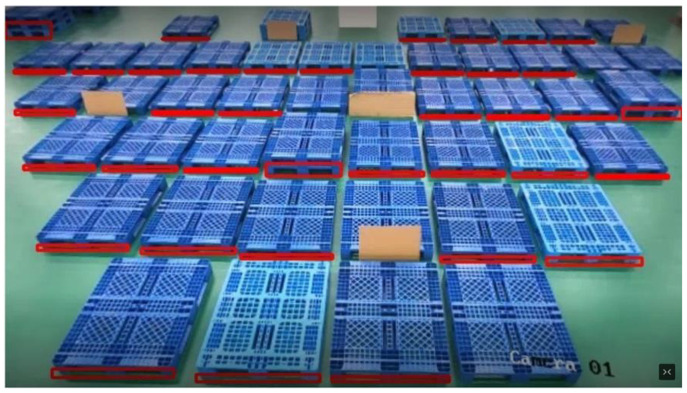
Recognition effect after using small target detection and CBAM. All pallets in the field of view, including multi-layer pallets and cargo pallets, are recognized.

**Figure 10 sensors-22-08437-f010:**
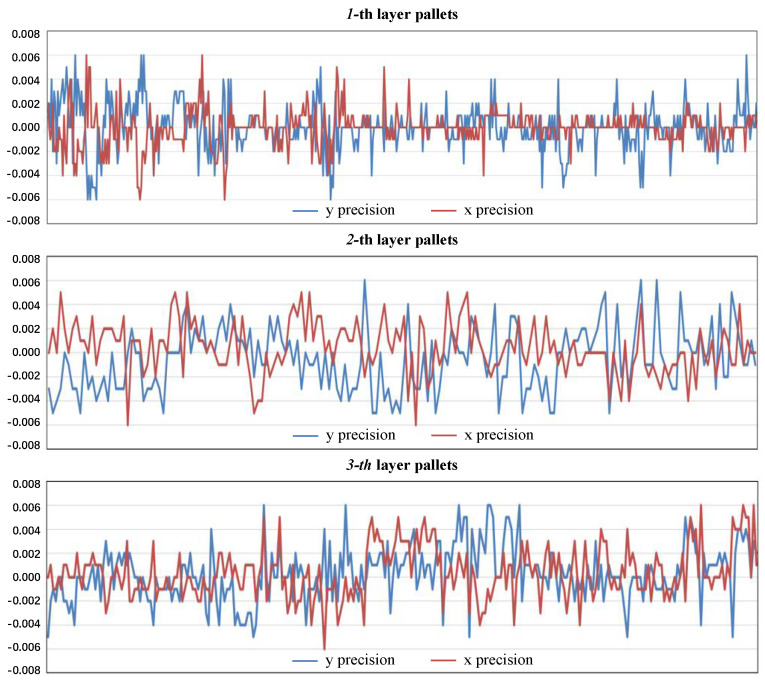
Results of pallet insertion experiment. 1-layer pallets test 550 times, 2-layer pallets test 180 times, 3-layer pallets test 260 times, and the maximum error of 6 mm.

**Table 1 sensors-22-08437-t001:** Comparison of pallet monitoring algorithms.

Algorithm	Method	Advantages	Disadvantages
Algorithm in [9]	photoelectric sensors	High recognition rate	High cost
Algorithm in [11]	image recognition	Low cost	Easily disturbed
Algorithm in [12]	Deep learning based on Yolov3	High stability	Low recognition rate for small targets
Our algorithm	Deep learning based on Yolov5	Low cost, high recognition rate	Algorithm is complicated

**Table 2 sensors-22-08437-t002:** Pallet monitoring reliability experiments.

Day	Number of Pallets	Recognition Accuracy (%)
1	72	99.82
2	70	99.76
3	72	99.68
4	68	99.86
5	71	99.69
6	74	99.58
7	69	99.71

**Table 3 sensors-22-08437-t003:** The effect comparison of three algorithms.

Algorithm	Range	Max Error (mm)	Mean Error (mm)	Std Error (mm)
Algorithm in [32]	1.5–4.5 m	30	n/a	n/a
Algorithm in [33]	2–4 m, 15°	10.5	7	3.9
Our algorithm	1–3 m, 15°	6	3.69	0.3

## Data Availability

Data will be made available upon request from the authors.

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
