# Peer review of "Deep Learning-Based Intelligent Forklift Cargo Accurate Transfer System"

_sensors, 2022, doi:10.3390/s22218437_

Round 1

Reviewer 1 Report

In the article “Deep Learning-Based Intelligent Forklift Cargo Accurate Transfer System”, the authors proposed an intelligent forklift cargo accurate transfer system consisting of pallet monitoring module, pallet positioning module and high precision control module. They proved its effectiveness and reliability by collecting a large amount of data.

The research topic is of importance and worthy of investigation. Overall the article is good, the methodology and procedure appear sound and the results are interesting. I recommend this paper for publication in Sensors after some minor corrections.

· It would further enhance the quality if the authors clearly mention the novelty of this study over the previous studies in abstract and conclusion sections.

·  I think there is no need for a separate literature review section, it is suggested to merge its contents in the introduction section instead. If this is the format of journal, then it is ok.

·  The resolution of images is low and need to be enhanced.

· I noticed several grammatical/ sentence structuring mistakes throughout the paper. It is suggested to thoroughly check the manuscript and correct such type of errors.

Reviewer 2 Report

Manuscript ID sensors - 1990402

Title: «Deep Learning-Based Intelligent Forklift Cargo Accurate Transfer System».

Overall, the topic of this paper is relevant, and the manuscript was good organized and written. Undoubtedly, the presented manuscript is relevant from a scientific and practical point of view. This study opens up defined prospects in this field of knowledge. Manuscript entitled "Deep Learning-Based Intelligent Forklift Cargo Accurate Transfer System" of interest to a highly ranked journal like "Sensors".

This manuscript includes the next harmonious structure:

1. Introduction (p. 1 – 2);

2. Related Works and Background (p. 2 – 4);

3. Intelligent forklift cargo precision transfer system (p. 4 – 9);

4. Experiment Results (p. 9 – 13);

5. Conclusions (p. 13).

Weakness and methodological inaccuracies are not detected. The figures and tables are appropriate, they properly show the data. So, they easy to interpret and understand by readers. References list is adequate and includes 34 titles.

The value of this work is significant, but I hope that next suggestions can help to improve the manuscript.

1) Please make clearer the aim of the study. In my opinion, general aim of the study is not formulated.

2) The titles of some figures are cluttered.

3) Indicate the directions of further research or improvements.

Reviewer 3 Report

1.This paper  proposes an intelligent forklift cargo precision transfer system to achieve better pallet docking accuracy and thus increase the recognition rate of the object to be identified. This technique allows an automatic verification of the transport of the pallet. As a result, an intelligent forklift comes into action after establishing a location recognition process.

2.The topic  of the article is about the implementation of an intelligent forklift accurate cargo transfer system, which is split into three essential parts: a pallet monitoring module, a pallet positioning module and finally a high precision control.

3.The novelty and originality of this research is mainly due to the practical aspect which has made it possible to prove significant efficiency and better reliability in collecting a large amount of data. Especially after implementing an image recognition system based on deep learning algorithm is used to calculate the precise position of the pallets. 

4.  The references are rich and adapted to the requirements of the subject.

5. The results and conclusions are consistent with the initial hypotheses. Moreover, the strategy adopted by the authors to implement a large place of practical tests has been fruitful. This is the strong point of the article.

Reviewer 4 Report

The authors propose an intelligent forklift cargo accurate transfer system, which consists of three main parts: a pallet monitoring module, a pallet positioning module, and high precision control module. For the whole system, we proved its effectiveness and reliability by actually collecting a large amount of data. Overall the paper is well-written and presents work on an interesting topic with a clear contribution. However, I have a few comments that could help to further improve the paper:
1- The beginning of the introduction should be redone with more emphasis on the general before highlighting the issue. 
2- Abstract and line 36, page 2, the abbreviation (RGBD, RGB, etc.) should be written in the long form at first use.
3- Wasn't there a way to use regression metrics such as MSE, MAE, R2, etc. to really evaluate the performance of the model and also compare it to previous work? 
4- Discussion of the results would have been a plus.

Reviewer 5 Report

In this manuscript, the authors reported that an intelligent forklift cargo accurate transfer system, which consists of three main parts: pallet monitoring module, pallet positioning module, and high precision control module. The authors claimed that the performance of the whole system is greatly higher than the existing common systems, and has been used in factories and warehousing environments on the ground. In overall, this manuscript is interesting but in order to consider publication, this work should be revised. The following comments should be addressed for the improvement of their manuscript.

Comment 1: The overall study aims for this study about the intelligent forklift cargo accurate transfer system need to be further clarified in detail as compared to current available / conventional type of forklift cargo.

Comment 2: The various recent reports and their research findings on the “deep learning-based intelligent forklift accurate cargo transfer system” for improving the resilience and accuracy of the system should be summarized into a table form and discussed for better understanding in term of benchmarking points with your research findings.

Comment 3: What is the role of “pallet positioning module and high precision control module” in improving the resilience and accuracy of the system need to be further discussed with fundamental support. 

Comment 4: The future direction and commercialization perspectives for this intelligent forklift cargo accurate transfer system can be further discussed in detail in the conclusion section.

Comment 5: The carefully English correction is necessary for the whole manuscript. Please check and revise accordingly.
